# Ensemble-averaged Rabi oscillations in a ferromagnetic CoFeB film

Amir Capua[1,2,3], Charles Rettner[1], See-Hun Yang[1], Timothy Phung[1] & Stuart S.P. Parkin[1,2]

Rabi oscillations describe the process whereby electromagnetic radiation interacts coherently with spin states in a non-equilibrium interaction. To date, Rabi oscillations have not been studied in one of the most common spin ensembles in nature: spins in ferromagnets. Here, using a combination of femtosecond laser pulses and microwave excitations, we report the classical analogue of Rabi oscillations in ensemble-averaged spins of a ferromagnet. The microwave stimuli are shown to extend the coherence-time resulting in resonant spin amplification. The results we present in a dense magnetic system are qualitatively similar to those reported previously in semiconductors which have five orders of magnitude fewer spins and which require resonant optical excitations to spin-polarize the ensemble. Our study is a step towards connecting concepts used in quantum processing with spin-transport effects in ferromagnets. For example, coherent control may become possible without the complications of driving an electromagnetic field but rather by using spin-polarized currents.

[1] IBM Research Division, Almaden Research Center, 650 Harry Road, San Jose, California 95120, USA. [2] Max Planck Institute for Microstructure Physics, Weinberg 2, D-06120 Halle (Saale), Germany. [3] Electrical Engineering Department, The Hebrew University of Jerusalem, The Edmond J. Safra Campus—Givat Ram, Jerusalem 9190401, Israel. Correspondence and requests for materials should be addressed to A.C. (email: amir.capua@mail.huji.ac.il) or to S.S.P.P. (email: stuart.parkin@us.ibm.com).

solated electron or nuclear spin states are ideal candidates for quantum information processing[1–5]. A practical gateway to this world is also provided by macroscopic quantum systems that are large cooperative ensembles[6]. Superfluids[7,8], superconductors[9,10] and ultracold dilute atomic vapours[11–13] are examples of such systems. In ferromagnets, macroscopic quantum behaviour asserts itself at low temperatures (mK) and/or small enough length scales (nanometre scale)[14–16]. In that limit, the observables of angular momentum obey the classical equations of motion. Hence, a great deal of insight into the quantum world is gained from studies of the classical ensemble-averaged analogues[17].

Spin ensembles in semiconductors have been considered in the context of quantum computing[18,19] for which resonant circularly polarized light is used to excite the superposition spin state at low temperatures. Nevertheless, dense ferromagnetic systems having five orders of magnitude more spins than semiconductors and which exhibit spontaneous spin-splitting at room temperature, are not currently considered suitable for such applications. Their spin states lack protection due to spin–spin and spin–lattice interactions[20]. While these may be overcome at low temperatures[14] or by careful engineering of their band structure[20], their coherent manipulation by non-adiabatic interactions remains unexplored, even in the classical limit.

The coherent manipulation of spin ensembles require operating in the non-adiabatic regime which pertains whenever the electromagnetic radiation and the state representing the ensemble are not in equilibrium, namely, when excessive energy is transferred back and forth between the ensemble and the oscillatory field before the steady precessional state is reached. In ferromagnets, the adiabatic interaction has been primarily explored using ferromagnetic resonance (FMR)[21,22] methods. Similar studies of the magnetic order have also been conducted by analysing the impulse response in the absence of the rotating field using the time-resolved magneto-optical Kerr effect (TR-MOKE)[23–27]. Despite the extensive studies of spin dynamics in ferromagnetic metals, little attention has been drawn to the non-adiabatic transitions. This non-equilibrium mode of operation can be accessed by either modifying the state of the radiation, for instance when the oscillatory field is turned on as in the application of $\pi$-pulses[28], or that of the magnetization, when the rotating field is already present and steady state conditions prevail.

Here we demonstrate the non-adiabatic regime in ensemble-averaged spins of a ferromagnet and show that a perturbation by an intense ultrashort optical pulse can initiate Rabi oscillations that are observable even in a dense ferromagnetic system. This is achieved in a ferromagnetic compound of CoFeB whose response is shown to be accurately described by Rabi's formula. The phase and amplitude responses reveal a frequency chirp which can be controlled by applying a static magnetic field. We find that the introduction of a transient torque in the form of a pulsed magnetic field occurring on a picosecond timescale better describes the interaction with the optical pulse. Our experiments further indicate that the microwave field induces coherence in the inhomogeneously broadened ensemble by selecting a subset of spins that are driven resonantly. Hence, the ensemble dephasing is suppressed and relaxation times that represent more closely those of individual spins result. Finally, in agreement with Gilbert's damping theory[29], we show that the intrinsic relaxation times can be tuned by proper choice of the external magnetic field[30] to initiate resonant spin amplification[19,31] that result in spin-mode locking[32] of the system. These effects show the potential of exploiting techniques developed for quantum coherent manipulation in ferromagnets.

## Results

**Sample details.** The sample studied was a $Co_{36}Fe_{44}B_{20}$ film having a thickness of 11 Å that was perpendicularly magnetized and grown by magnetron sputtering. The effective anisotropy field, $\mu_0 H_{Keff}$, was measured to be $\sim 140$ mT, with $\mu_0$ being the magnetic permeability. From TR-MOKE measurements of the free induction decay responses (Fig. 1a) a Gilbert damping constant, $\alpha$, of 0.023 and an inhomogeneous distribution of the effective anisotropy field, $\mu_0 \Delta H_{Keff}$, of 17.5 mT were determined[33]. We describe here the dynamical processes in the ferromagnet using $\alpha$ and $\Delta H_{Keff}$ rather than using the spin–lattice, and transverse spin polarization decay times[34,35], $T_1$ and $T_2^*$, respectively, which are commonly used to describe quantum coherent phenomena, due to the physical origins of the Gilbert damping[29] (Supplementary Note 1).

**Rabi oscillations in a ferromagnet.** The concept of our experiment is presented in Fig. 1b. A microwave field is used to drive spin precessions in the film which are then perturbed by a femtosecond optical pulse that is phase-locked to the microwave signal. The temporal recovery is recorded by a weak optical probe pulse as a function of a pump-probe delay time using the MOKE detection scheme. Example of the measured responses for three values of applied magnetic field, $H_0$, are presented in Fig. 1c–e. A distinct envelope is seen to modulate the carrier signal. This envelope exhibits a systematic behaviour; the time of its minimum increases as $H_0$ approaches the resonance field, $H_{res}$. These traces do not stem from spin wave interference generated by the microwave and by the optical pump as we show in the following.

The responses for a complete set of $H_0$ fields, are illustrated in Fig. 1f. At $H_0 < H_{res}$, the shift in time of the minima is seen clearly and forms a valley. At $H_0 > H_{res}$, a maximum is formed instead, making the response asymmetric. Plotting in addition the theoretical prediction for these signatures as given by the generalized Rabi formula for the frequency of the nutations:

$$\Omega_R^G = \sqrt{(\gamma \mu_0 H_0 - \omega_{RF})^2 + (\gamma \mu_0 h_{RF})^2} \qquad (1)$$

results in a very good agreement with the measured features. In equation (1) $\gamma$, $h_{RF}$ and $\omega_{RF}$ are the gyromagnetic ratio, microwave amplitude and microwave angular frequency, respectively. The equation was derived under the assumption that the magnetocrystalline and demagnetization fields are small compared to the external field. The apparent agreement indicates that the ensemble-averaged Rabi oscillations are observable in the ferromagnetic film and implies that the well-established methods and techniques used in coherent quantum processing can be applied on ferromagnetic materials and vice versa[36].

Regarding the coherence times of the interaction, while the Rabi cycles we record occur on timescales of few hundreds of picoseconds depending on the detuning from resonance, the effective coherence time is $\sim 500$ ps which consists of the intrinsic decay time of $\sim 1.15$ ns and the dephasing lifetime due to the inhomogeneous broadening of $\sim 0.95$ ns (Methods section). Hence, the coherence times are longer than the Rabi cycle and allow its observation.

**Phase responses.** The phase responses which are generally less accessible in experiments that address the coherent light-matter interactions are also measurable in our experiment. We analyse them by plotting the data set of Fig. 1f as a two-dimensional contour plot (Fig. 2a). Before the pump pulse arrives, a net phase shift of $\sim 0.75\ \pi$ is measured across the resonance (Fig. 2b) as expected. Surprisingly, at times (pump-probe delays) well after the perturbation, a phase shift of $\sim 2.75\ \pi$ is observed as $H_0$ is

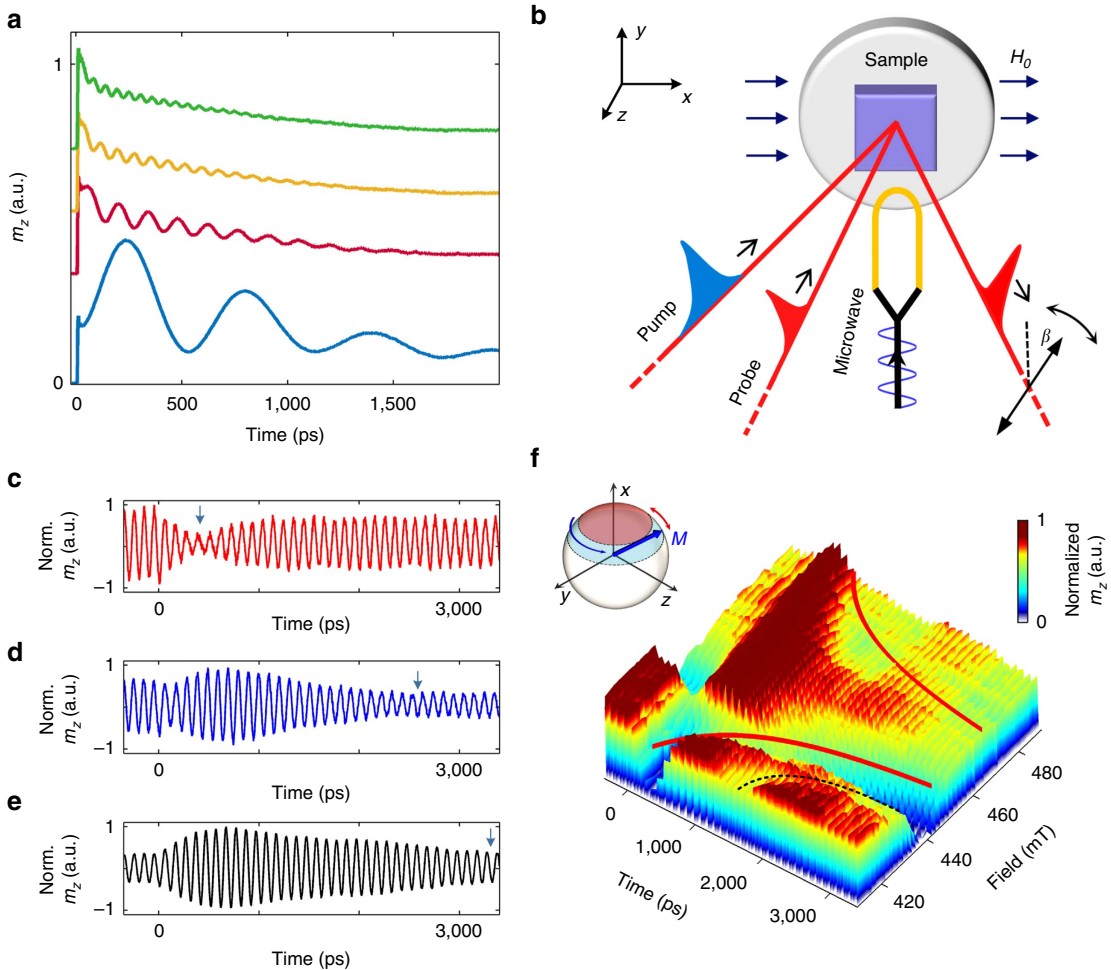

**Figure 1 | Temporal responses in the non-adiabatic regime.** (**a**) Free induction decay responses of the TR-MOKE experiment for applied magnetic fields of 100 mT (blue), 300 mT (Red), 500 mT (yellow) and 700 mT (green). Traces are shifted along the $y$ axis for clarity. (**b**) Schematic of the experimental set-up. A femtosecond pulse laser is phase-locked to a microwave oscillator. External magnetic field is applied in the sample plane causing precessions about the $x$ axis while the out-of-plane component of the magnetization, $m_z$, is detected in a polar-MOKE configuration. The magnetic film was patterned into a square island and the microwave signal was transmitted by a shorted Au microwire. $\beta$ represents the angle of polarization rotation. (**c-e**) Temporal responses of the pump-probe FMR measurement at 10 GHz and $H_0$ values of 422 mT (**c**), 434 mT (**d**), and 443 mT (**e**) and microwave field amplitude of ∼0.8 mT. The pump pulse arrives at $t = 0$ ps. The resonance field at this frequency is $\mu_0 H_{res} \sim 450$ mT. Each trace is normalized to the peak value. Arrows indicate the envelope minimum. (**f**) Measured temporal responses at 10 GHz for a complete range of applied fields. Each trace is normalized individually to the peak value. The solid red lines were plotted using the Rabi formula. Also the second Rabi oscillation is readily seen (black dashed line). Inset illustrates the trajectory of the magnetization vector, **M**, on the sphere of constant saturated magnetization. 10 GHz magnetization precessions take place about the $\hat{x}$ axis while the slow variations in the $\hat{x}$, $\hat{y}$ or $\hat{z}$ component of **M** obey Rabi's formulae.

varied (Fig. 2c). The instantaneous frequency profiles (Fig. 2d) explain the behaviour. Apart from the sharp transient at $t = 0$, the temporal instantaneous frequency responses reveal a negative, zero, and positive chirp profiles corresponding to $H_0 < H_{res}$, $H_0 = H_{res}$ and $H_0 > H_{res}$, respectively. At long delays, the instantaneous frequency recovers to the driving frequency, independent of $H_0$. This behaviour is explained by recalling that Rabi oscillations can be represented as a beating of the natural transient decaying precessional response of the system at the angular frequency of $\gamma \mu_0 H_0$ with the steady state response at $\omega_{RF}$ (ref. 37) as in the solution to the problem of the driven damped harmonic oscillator. Hence, when $\gamma \mu_0 H_0 < \omega_{RF}$, a negative chirp initially takes place which recovers to $\omega_{RF}$. The same explanation holds also for other $H_0$ values and refutes the seemingly intuitive picture of spin wave interference. Therefore, variation of $H_0$ provides a means of controlling the effective pulse area (the time-integrated Rabi frequency) of the microwave radiation.

The phase information in our measurement brings an additional point of view on the interaction of the optical pump with the ferromagnet, a topic of much debate and controversy.[26,38,39] It is generally understood that the optical pump increases the lattice temperature thereby reducing temporarily $H_{Keff}$ and the magnetization saturation, $M_s$. (ref. 23) Incorporating these thermal effects in a macrospin numerical model of the Landau–Lifshitz–Gilbert equation was not sufficient to reproduce the measured traces (Supplementary Note 2). However, introducing a transient effective torque that is presumably induced by the optical field resulted in a good agreement with the measured field-dependent phase response near $t = 0$ (Supplementary Fig. 2). This torque had the form of a 3 ps pulsed magnetic field of 60 mT which lied along the $\hat{y}$ direction (axis indicated in Fig. 1b). Though its existence or uniqueness cannot be completely verified, it is consistent with previous suggestions that a non-thermal optically induced

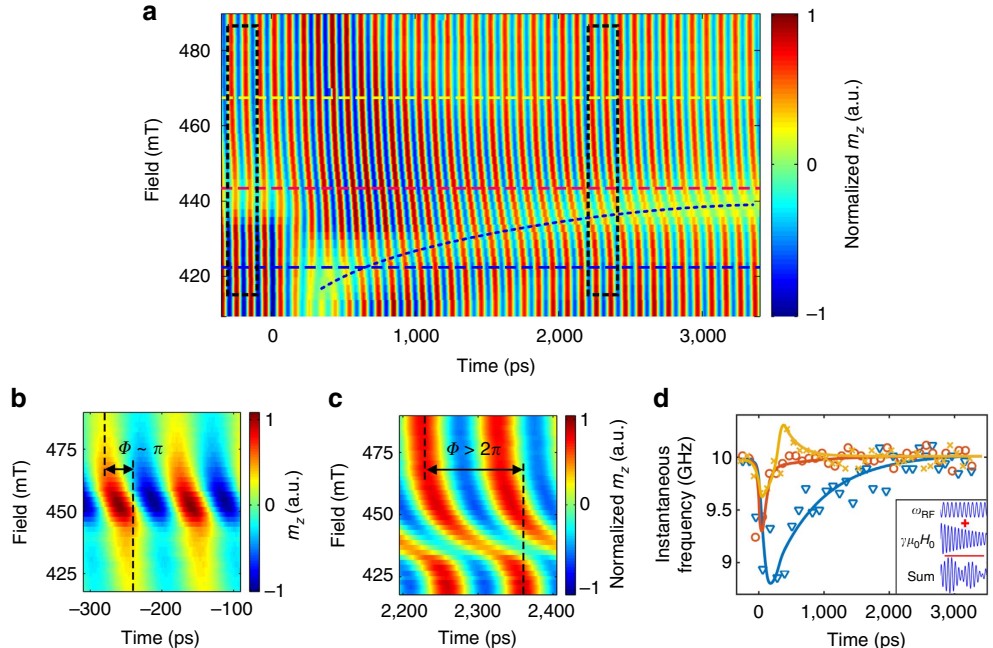

**Figure 2 | Phase response. (a)** Dependence of the phase responses on the applied field at 10 GHz. Data set of Fig. 1f is presented in a two-dimensional contour plot to represent the phase information. Each temporal response was normalized individually. Blue curved guiding line indicates the location of the valley in Fig. 1f. **(b)** Phase response before the perturbation. The figure presents a close-up of the black dashed area of **a** for times between $-300$ and $-100$ ps. An overall phase shift of $\sim 0.75\,\pi$ is measured across the resonance as indicated by the vertical black dashed lines representing the phase front at highest and lowest applied fields. Data are not normalized. **(c)** Phase response at long delays corresponding to black dashed area in **a** for times between 2,200 and 2,400 ps. Data are presented in normalized units. The measured net phase shift across the resonance is $\sim 2.75\,\pi$. Black dashed lines represent the phase front at highest and lowest applied fields. **(d)** Instantaneous frequency profiles at $\mu_0 H_0$ values of 424 mT (blue), 444 mT (red) and 468 mT (yellow) corresponding the blue, red and yellow dashed lines of **a**, respectively. Inset illustrates the decomposition of the total response to the steady state response of the system at $\omega_{RF}$ and the natural response at the angular frequency of $\gamma\mu_0 H_0$. The same reasoning also accounts for the asymmetry seen in the responses of Fig. 1f.

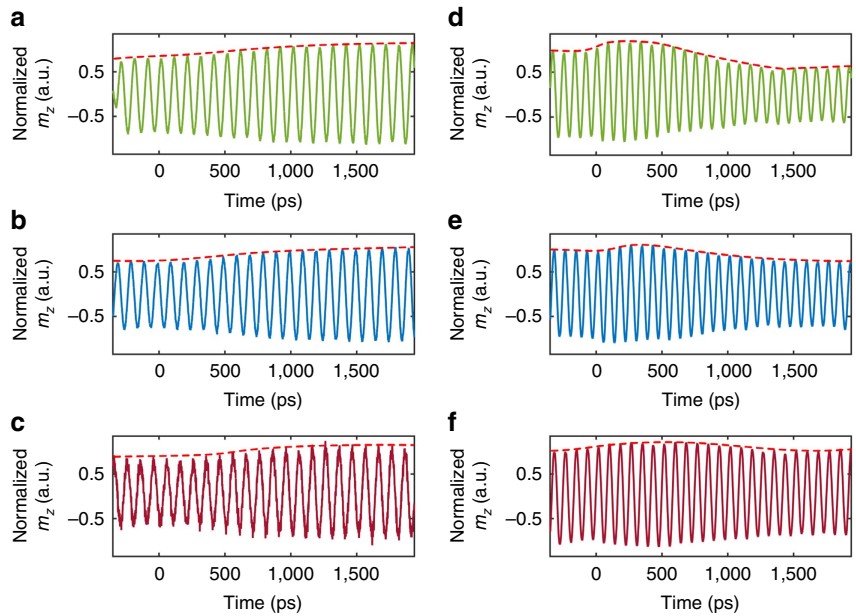

**Figure 3 | Microwave power dependence of the Rabi frequency. (a–c)** Temporal responses for the CoFeB sample for microwave field amplitudes of 7.5 mT **(a)**, 1.3 mT **(b)** and 0.25 mT **(c)**. Responses were measured at a frequency of 10 GHz and $\mu_0 H_0 = 446$ mT. The envelopes (red dashed lines) exhibit a similar temporal form independent of the microwave amplitude. **(d–f)** Temporal responses of a 4 nm-thick molecular beam epitaxy (MBE) grown single-crystal Fe sample at microwave field amplitudes of 7.5 mT **(d)**, 4.7 mT **(e)** and 2.3 mT **(f)**. Responses were measured at a frequency of 12 GHz and $\mu_0 H_0 = 143$ mT. In contrast to the sputter deposited CoFeB film, the envelope exhibits a clear dependence on the microwave amplitude. Measurements in **a–f** were carried out at the resonance conditions. Guiding dashed lines indicate the envelopes.

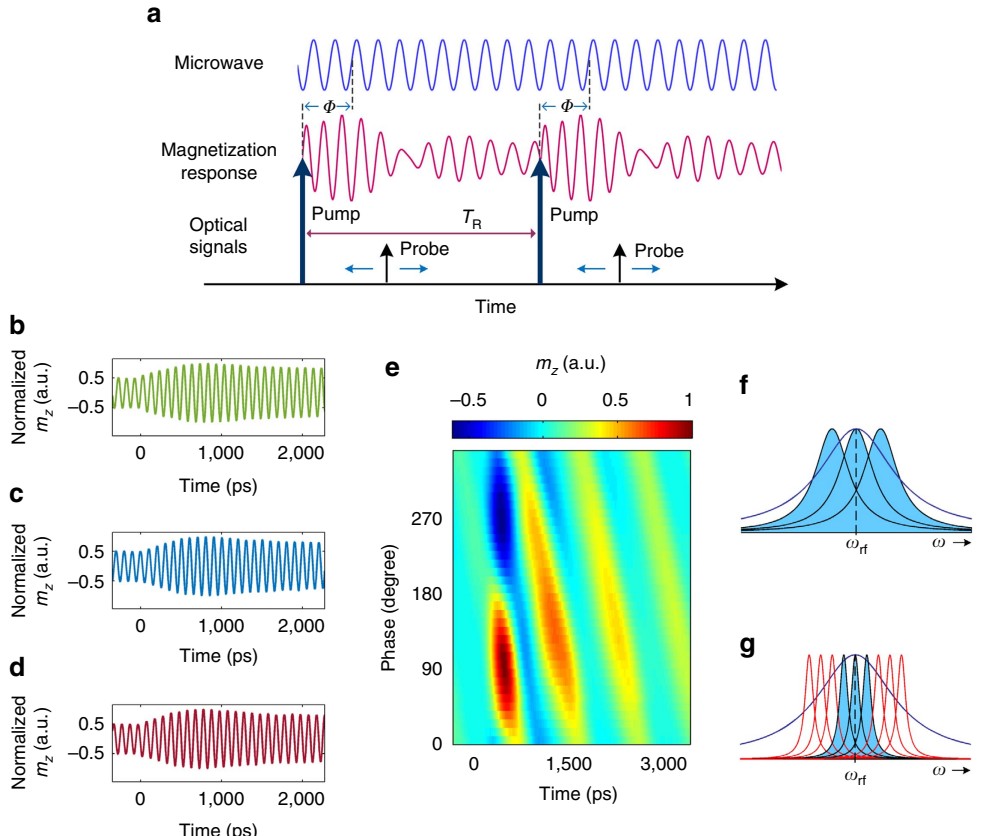

**Figure 4 | Microwave spin-subgroup selection and resonant spin amplification.** (**a**) Schematic arrangement of the signals in time. (**b**–**d**) Dependence of the temporal response on the relative phase between the optical pump pulse and the microwave signal, $\Phi$, for short intrinsic lifetime, $\tau_{int}$. Data presented for a frequency of 10 GHz and $\mu_0 H_0 = 450$ mT and relative phases of 0° (**b**), 90° (**c**) and 180° (**d**). Variation of $\Phi$ has no effect on the envelope; the carrier signal shifts within the same envelope. (**e**) Dependence of temporal response on $\Phi$ for long intrinsic lifetime, $\tau_{int}$. Data is presented for a frequency of 1 GHz and $\mu_0 H_0 = 90$ mT. For $\Phi = 90°$, constructive interference results in a sharp pulsation of the magnetization. Likewise, for additional 180°, at $\Phi = 270°$, pulsations of opposite polarity are generated. When the phase is tuned to $\Phi = 0°$ and $\Phi = 180°$, destructive interference takes place and no pulsations are observed. (**f**) Illustration of the inhomogeneous broadening at 10 GHz corresponding to **b**–**d**. (**g**) Illustration of the inhomogeneous broadening at 1 GHz corresponding to **e**. In **f**,**g**, blue solid line indicates the total effective resonance line which includes contributions of the inhomogeneous broadening. Blue shaded resonances indicate the subgroups that are selected by the microwave. Red solid lines indicate the subgroups that are not interacting with the microwave signal. Data are shown for the CoFeB sample. The linewidths presented were determined from the experimental data.

magnetic field may exist[38]. Possible sources for such field include demagnetization and anisotropy change induced field[40,41], photo-induced anisotropy[42], the inverse Faraday effect[42–45], as well as magnetic circular dichroism[46] and are further discussed in Supplementary Note 3. It is possible that at lower optical pump intensities the Landau–Lifshitz–Gilbert equation may solely account for the experimental results. Furthermore, it is worth noting that although our macrospin model lacks spatial resolution, it reproduces the experimental results. Hence, spin waves are not included in the calculation and cannot account for our experimental observations.

**Microwave amplitude dependent Rabi oscillations.** Another means of controlling the effective pulse area is by modifying the amplitude of the microwave field. This effect is most readily seen under resonance conditions for which the generalized Rabi frequency simplifies to $\Omega_r = \gamma \mu_0 h_{RF}$. The measured results are shown in Fig. 3a–c. Surprisingly, no dependence of the envelope on the microwave amplitude is revealed. This apparent discrepancy is resolved by considering the contributions to equation (1). The maximal applied microwave field amplitude was $\mu_0 h_{RF} \sim 7.5$ mT while the inhomogeneous linewidth broadening at 10 GHz, as

derived from the value of $\Delta H_{Keff}$ (ref. 33), is 10.5 mT meaning that the off-resonant detuning term in equation (1) is dominating the response regardless of the microwave power. When the measurement was repeated on a single-crystal Fe sample which was epitaxially grown and hence exhibited a vanishingly small inhomogeneous broadening, the predicted increase in the Rabi frequency according to Rabi's formula was verified (Fig. 3d–f).

**Resonant spin amplification.** Next, we turn to show that the train of optical pulses can be utilized to lock the phases of the spins constituting the ensemble by the process of resonant spin amplification[19,31]. This mode of operation requires that coherence persist for a duration longer than the laser repetition time, $T_R$ (Fig. 4a). As follows from Gilbert's theory for damping, the rate of transfer of spin angular momentum to the lattice, $1/\tau_{int}$, can be controlled by the magnitude of $H_0$ (ref. 33). Accordingly, interference effects between subsequent responses are expected at low $H_0$ values. The nature of the interference will then depend on the relative phase, $\Phi$, between the optical pump pulse and the microwave signal (Fig. 4a).

Measured responses as a function of $\Phi$ are presented in Fig. 4b–e. At high magnetic field ($\mu_0 H_0 = 450$ mT) and short $\tau_{int}$

($\sim 1.1$ ns) compared to $T_R$ of 12.5 ns, the interaction of each pump pulse within the train of pulses can be regarded as an isolated event (Fig. 4b–d). In contrast, for low external magnetic fields ($\mu_0 H_0 = 90$ mT) and correspondingly long $\tau_{int}$ ($\sim 5$ ns), interference occurs and the moment at which the optical pulse is sent becomes critical (Fig. 4e). The observed pulsations of the magnetization indicate that the spins within the ensemble have become synchronized, that is, mode-locking takes place[32].

In addition to the intrinsic relaxation, the decay of the transient response is governed also by dephasing of the inhomogeneously broadened ensemble so that the effective decay time of the response, $\tau_{eff}$, is given by: $1/\tau_{eff} = 1/\tau_{int} + 1/\tau_{IH}$ where $\tau_{IH}$ represents the ensemble dephasing[33]. While a fundamentally different dependence on $\Phi$ is observed in the two regimes of Fig. 4b–e, the inhomogeneous broadening causes $\tau_{eff}$ to be very similar in both cases and corresponds to $\sim 0.51$ and $\sim 0.49$ ns, respectively. This fact shows that the long intrinsic relaxation time, $\tau_{int}$, in the case of low $H_0$ (Fig. 4e) can be sensed despite the significant ensemble dephasing. By use of the relations $\Delta \omega_{int} = 2/\tau_{int}$ and $\Delta \omega_{IH} = 2/\tau_{IH}$ for the intrinsic resonance linewidth and inhomogeneous broadening, respectively, $\Delta \omega_{int} \approx 1.75$ rad GHz and $\Delta \omega_{IH} \approx 2.15$ rad GHz were extracted for $\mu_0 H_0 = 450$ mT, while $\Delta \omega_{int} \approx 0.43$ rad GHz and $\Delta \omega_{IH} \approx 3.66$ rad GHz were found for $\mu_0 H_0 = 90$ mT (Fig. 4f,g). In contrast to the high $H_0$ case, at low $H_0$ only a subset of spins which exhibit long $\tau_{int}$ are interacting while the interaction with the off-resonant spins is suppressed, namely, the microwave induces coherence in the ensemble. Hence, the inhomogeneity is effectively overcome and $\tau_{eff}$ extends towards its upper limit of $\tau_{int}$ that describe the individual spins (see Supplementary Note 4 for more information).

**Conclusions**. In summary, in this work we report the non-adiabatic regime in dense ferromagnetic metals using a pump-probe FMR type measurement. Extensions of the present work will have wide applicability to the exploration of more complex coherent phenomena in ferromagnets such as Ramsey interference, self-induced transparency, Hahn spin echo and slow light applications with magnetostatic waves, and of more complex magnetic systems such as magnetic systems with multiple components. In particular, the reported observations allow to make the connection between concepts that are used in quantum information processing and apply them on the atomically engineered ferromagnetic systems. For example, the electromagnetic radiation required to manipulate the magnetic moments in the non-adiabatic interaction can be replaced by spin-transfer torque from spin currents generated by the spin Hall effect to cause similar precessions. In such case Rabi nutations are made possible without the complications of a driving electromagnetic field.

## Methods
**Sample preparation.** The CoFeB film was prepared on thermally oxidized Si(100) substrate and consisted of the following structure, starting from the substrate side: SiO$_2$ (250)/Ta (100)/CoFeB (11)/MgO (11)/Ta (30) (numbers are in nominal thicknesses in angstroms). The MgO layer was deposited by RF sputtering. The sample was annealed at a temperature of 275 °C for 30 min while applying a 1 T field in the out-of-plane direction. The polycrystalline CoFeB thin film has a body centred cubic crystal structure that is highly non-textured. The interfaces and boundaries of the crystallites naturally further reduce the symmetries. In-plane and out-of-plane magnetization loops are shown in Supplementary Fig. 5. The single crystalline 40 Å-thick Fe film was grown on a MgO(100) substrate using molecular beam epitaxy.

For the pump-probe FMR measurement the samples were patterned to a magnetic island of $20 \times 20$ $\mu$m$^2$ using electron-beam lithography. A shorted Au microwire was having lateral dimensions of $5 \times 20$ $\mu$m$^2$ and a thickness of 100 nm was patterned at a distance of 1 $\mu$m away from the island by lift-off to drive the microwave signal.

**Ferromagnetic resonance pump-probe measurement.** A Ti:Sapphire oscillator emitting $\sim 70$ fs linearly polarized pulses at 800 nm having energy of $\sim 5$ nJ per pulse was used for the optical measurements. The beam was focused to a spot size of $\sim 10$ $\mu$m. The pump beam was applied at an incident angle of $\sim 22°$ measured from the normal to the sample plane. The probe pulses were attenuated by 20 dB relative to the pump. The timing jitter between the optical pump and the microwave signal was measured to be smaller than 1 ps. All measurements were carried out at room temperature. The maximum microwave power applied was 1 W and corresponded to an amplitude of $\sim 7.5$ mT.

In the pump-probe FMR measurements, a double lock-in detection scheme was used for which the microwave signal was modulated at 50 KHz and the optical probe at 1 KHz. To exert sufficient torque by the optical pump, the external magnetic field was applied at an angle of 4° away from the sample plane. The same arrangement was used also in the TR-MOKE measurements. The dimensions of the Au microwire were $5 \times 20$ $\mu$m$^2$ while its thickness was 100 nm. The signal was launched into the microwire from a 50 ohm microstrip. Care was taken that the bonding to the microwire was significantly shorter than the quarter radio frequency (RF) wavelength so that the microwire can be considered as a lumped short circuit element which in principle reflects all energy besides the radiation losses. To verify our considerations the microwire was simulated using the high frequency structural simulator.

FMR measurement without the presence of the optical pump were carried out also at other RF powers. A heating effect would have changed the magnetization saturation, $M_s$, which in turn would have modified $H_{Keff}$. From these measured responses we could not identify any heating effects. The reflected energy was absorbed in the third port of a microwave isolator which indeed heated up to some extent.

**Extraction of decay times from TR-MOKE measurements.** Extraction of the ensemble dephasing times and the intrinsic spin relaxation times from TR-MOKE measurements was based on the analysis presented in ref. 31. Accordingly, $\alpha$ and $\Delta H_{Keff}$ were obtained by fitting the measured effective linewidths, $\Delta \omega_{eff}$, with the equation:

$$\Delta \omega_{eff} = \alpha \gamma \mu_0 (2H_0 - H_{Keff}) + \frac{\gamma H_0}{2\sqrt{H_0^2 - H_0 H_{Keff}}} \mu_0 \Delta H_{Keff} \quad \text{for} \quad H_0 > H_{Keff}$$

$$\Delta \omega_{eff} = \alpha \gamma \mu_0 H_0 \left( \frac{2H_{Keff}}{H_0} - \frac{H_0}{H_{Keff}} \right) + \frac{\gamma H_{Keff}}{\sqrt{H_{Keff}^2 - H_0^2}} \mu_0 \Delta H_{Keff} \quad \text{for} \quad H_0 < H_{Keff}$$

(2)

Here, $\Delta \omega_{eff} = 2/\tau_{eff}$, and $\tau_{eff}$ is the overall decay time of the precessional motion as measured in the TR-MOKE experiment. This analysis is valid for $H_0$ much larger or much smaller than $H_{Keff}$. The first terms of the equation represent the intrinsic linewidth $\Delta \omega_{int}$ while the second terms represent the inhomogeneous broadening $\Delta \omega_{IH}$. For $H_0 \sim H_{Keff}$, as in the case where $\mu_0 H_0 = 90$ mT, $\tau_{int}$ and $\tau_{IH}$ were extracted numerically[33].

**Data availability.** The data that support the findings of this study are available from the corresponding author on request.

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

## Acknowledgements

We thank Dr Dan Rugar, Dr Chris Lutz and Dr John Mamin for fruitful discussions, and Chris Lada for expert technical assistance. A.C. thanks the Viterbi Foundation and the Feder Family Foundation for supporting this research.

## Author contributions

A.C. conceived the experiments and carried them out. S.-H.Y. grew the samples. C.R. patterned the devices. A.C., S.-H.Y., T.P., C.R. and S.S.P.P. analysed the data and co-wrote the paper.

## Additional information

**Competing interests:** The authors declare no competing financial interests.

