## [Peer Review File · Nature Communications]

Reviewers' comments:

Reviewer #3 (Remarks to the Author):

I believe the author has addressed all of my objections and questions, modifying their manuscript accordingly. I also believe the authors addressed the other referees' points satisfactorily. Hence I recommend publication.

Reviewer #4 (Remarks to the Author):

The manuscript reports about an experimental study of the laser-induced magnetization dynamics in CoFeB film under microwave pumping. The experiment is quite original. The results are of high quality and new. However, I am very much confused by the way the authors present the data. To me, there is no need to describe the observed phenomenon as Rabi oscillations. The mechanisms of the excitation of the oscillations and the role of the femtosecond laser pulse are hardly explained in the manuscript. As a result, the manuscript is very difficult to read.

Regarding the interpretation suggested by the authors I have three serious concerns

1) In line with Referee 1, I think that the phenomenon observed by the authors can be (and thus should be) described in terms of classical physics. In the experiment described in this paper, classical spin waves induced by microwave pumping interfere with classical spin waves induced by laser pulse. The result of the interference the authors try to describe quantum-mechanically as Rabi oscillations. Note that real Rabi oscillations would have been observed under action of microwave pumping only. The experiment performed by the authors is clearly different (additional repetitive laser pumping is present) and thus the observed effect is also different from Rabi oscillations.

2) There is some analogy between the phenomenon observed by the authors and Rabi oscillations. I can believe that the Rabi formula can explain the beatings observed by the authors. However, mathematical description of the observed beatings is clearly different from those of the Rabi oscillations. To observe the Rabi oscillations an ensemble of quantum two-level systems should maintain coherence on a time-scale longer than the period of the Rabi oscillations. Mathematically it is quite different from the described forced oscillations driven by microwave pumping. Hence I must conclude that also in theory the observed phenomenon is quite different from Rabi oscillations.

3) Compared to conventional experiments on Rabi oscillations, the experiments described here require additional laser pumping. It is claimed that the laser pulse acts as a pulse of a magnetic field. Why? What was the direction of the field? What are physical quantities that define the strength and the direction of this laser-induced field? Is it realistic to expect such a strength and duration of the laser-induced magnetic field? The authors hardly explain these issues. The physics of the effect of the laser pulse on the magnetic medium leaves a lot of questions raising doubts about the validity of the suggested model.

Reviewer #5 (Remarks to the Author):

The authors study Rabi oscillations in a thin-film ferromagnet by driving magnetization precession with a microwave field and perturbing the system with femtosecond laser pulses. They show that the perturbation leads to a modulation of the magnetization precession amplitude. The characteristic frequency of this modulation is found to be consistent with a generalized Rabi formula. Further, they investigate how the laser pulses influence the phase of the oscillating signal (as function of the applied magnetic field) and how the Rabi frequency depends on the magnitude of the microwave field. They also study interference effects between subsequent responses to a

train of laser pulses.

The topic of this work is innovative and the results are of broad interest. I also believe that the presented data are technically sound. The observed oscillations (with a signature consistent with Rabi oscillations) have not been reported before in ferromagnets and the results give valuable insights into the interaction between laser pulses and magnetization (e.g., an improved understanding of optical magnetization switching and how to model the interaction phenomenologically). In particular, I believe physicists working within spintronics will find this article useful for their future works. However, I think the authors overstretch the link to quantum computation, because they consider a classical system and a large ensemble of spins (not a quantum mechanical two-level system). Another weak point of the paper is its presentation. It is difficult to get a good understanding of this work without reading the article a couple of times. For instance, I miss a better introduction that gives a simple physical picture of the effect.

In conclusion, the results are novel, technically sound, and will be important to scientists working within spintronics. Therefore, I believe this paper satisfies the criteria set for being publishable in Nature Communications and recommend publication after the authors have improved the introduction with a simple physical picture of the observed Rabi oscillations.

Kjetil M. D. Hals

Reviewer #3 (Remarks to the Author):

I believe the author has addressed all of my objections and questions, modifying their manuscript accordingly. I also believe the authors addressed the other referees' points satisfactorily. Hence I recommend publication.

We kindly thank the referee for his/her comments and for reviewing our manuscript.

Reviewer #4 (Remarks to the Author):

The manuscript reports about an experimental study of the laser-induced magnetization dynamics in CoFeB film under microwave pumping. The experiment is quite original. The results are of high quality and new. However, I am very much confused by the way the authors present the data. To me, there is no need to describe the observed phenomenon as Rabi oscillations. The mechanisms of the excitation of the oscillations and the role of the femtosecond laser pulse are hardly explained in the manuscript. As a result, the manuscript is very difficult to read.

Regarding the interpretation suggested by the authors I have three serious concerns

1) In line with Referee 1, I think that the phenomenon observed by the authors can be (and thus should be) described in terms of classical physics. In the experiment described in this paper, classical spin waves induced by microwave pumping interfere with classical spin waves induced by laser pulse. The result of the interference the authors try to describe quantum-mechanically as Rabi oscillations. Note that real Rabi oscillations would have been observed under action of microwave pumping only. The experiment performed by the authors is clearly different (additional repetitive laser pumping is present) and thus the observed effect is also different from Rabi oscillations.

We thank the referee for the careful reading of our manuscript and for his/her comments. Though intuitive, the explanation that the interaction stems from interference of classical spin waves that were generated by the microwave and by the pump laser (PRL 110, 097201) is incorrect and cannot account for the observed results. For example, it cannot explain the observed chirp profile seen in Fig. 2D and even less so its field dependence. The responses are however describable by a beating effect of the natural decaying precessional response at the frequency of $\mu_0 H_0$ with the driven steady state response at the frequency of ω_f and is the central idea behind Rabi oscillations. This is the origin of the field dependence we observe for example.

Furthermore, the numerical model we use was calculated in the macrospin limit and had no spatial resolution. Hence, it did not account for spin waves. Nevertheless, it reproduced the experimental results reliably (Fig. S3).

The interference process suggested by the reviewer does not require the non-adiabatic regime whereas the essence of the interaction we report is the non-adiabatic interaction for which energy is transferred back and forth between the magnetic media and the electromagnetic radiation and is hence fundamentally different than classical spin wave interference.

The necessary condition to observing Rabi oscillations is that the non-adiabatic regime prevail regardless of how it was reached. In our experiments this is reached by perturbing the microwave driven ferromagnet away from its steady state using the optical pulses. However, the non-adiabatic regime can be equally achieved by turning on the microwave field without the optical pulse, as the referee suggests. This may be an important topic for a future study; exploring the appearance of Rabi oscillations in the frequency domain using the standard ferromagnetic resonance (FMR) method together with pulsed microwave signals.

Having said that, and in addition to the basic fact that Rabi's formula agrees very well with the experimental results, we can satisfactorily conclude that the observed phenomena do not stem from classical spin waves but are rather described by Rabi's theory.

To address the referee's concerns, we have revised the introduction to what we believe is now more easily understandable. Furthermore, phrases that might still have caused confusion between the classical and quantum pictures have been removed. In addition, throughout the text, care was taken to clarify that the effects we report do not stem from classical spin wave interference.

2) There is some analogy between the phenomenon observed by the authors and Rabi oscillations. I can believe that the Rabi formula can explain the beatings observed by the authors. However, mathematical description of the observed beatings is clearly different from those of the Rabi oscillations. To observe the Rabi oscillations an ensemble of quantum two-level systems should maintain coherence on a time-scale longer than the period of the Rabi oscillations. Mathematically it is quite different from the described forced oscillations driven by microwave pumping. Hence I must conclude that also in theory the observed phenomenon is quite different from Rabi oscillations.

We beg to differ with the referee's opinion. Rabi's formula was derived for a spin system in a rotating electromagnetic field in the non-adiabatic regime, namely, before energy balance between the electromagnetic radiation and spin system was reached rather than being in a dynamical steady state. Commonly, Rabi oscillations are demonstrated when the electromagnetic radiation is turned on, for example in the application on a " π -pulse". However, as we show here, it is possible also to reach the non-adiabatic regime even for a system that has already reached its dynamical steady state if the state of the spins are perturbed rather than the field. The perturbation in our case is carried out using the optical pulse. The difference from the " π -pulse" case is merely in the initial conditions of the system which Rabi's formalism is obviously independent of. Hence, Rabi's formula can be adopted as is. The only adaptation required is in the gyromagnetic ratio, γ , which corresponds to the nuclear spin in Rabi's works whereas in ferromagnets it corresponds to the electron spin.

Indeed, in order to observe Rabi oscillations the ensemble coherence should be preserved on times longer than the Rabi cycle. The well-known T_1 and T_2 times are commonly used to

describe in a phenomenological manner the coherence times and represent the intrinsic longitudinal relaxation time and the transverse dephasing time, respectively. This description was originally presented in the Bloch-Bloembergen formalism (*Phys. Rev.* **93**, 72 & *Phys. Rev.* **73**, 678). Later Gilbert derived a more rigorous formalism that was based on Raleigh's dissipation function and was suitable for ferromagnets (summarized in *IEEE Transactions on Magnetics* **40**, 3443) and was therefore naturally adopted by the magnetism community. In our work we adopt Gilbert's formalism, by using the Gilbert damping coefficient, α , which can be considered as a combination of T_1 and T_2 to some extent, while the inhomogeneous broadening stems from a spatial variation of the anisotropy field, H_{Keff} , via the parameter ΔH_{Keff} (see Shaw et al. *APL* **105**, 062406, Iihama et al., *PRB* **89**, 174416, for example). Hence, in that respect as well, the theory we use is not different from the theory describing Rabi oscillations. In fact, it should be interesting to borrow the well-established theories in magnetism to other fields of physics such as NV centers (Neumann, *Science* **320**, 1326), excited electron spins in semiconductors (Greilich, *PRL* **96**, 227401), superconducting qubits (Simmonds et al. *PRL* **93**, 077003) etc.

As to the particular timescales of our experiments, for the CoFeB sample presented in Fig. 1D, for example, the Rabi cycles we record are on the order of magnitude of several hundreds of picoseconds depending on the detuning from resonance. On the other hand the effective coherence time is ~ 500 ps out of which the intrinsic lifetime is ~ 1.15 ns and dephasing lifetime due to the inhomogeneous broadening is ~ 0.95 ns (see supplemental materials, and paragraph second to last). Hence, we can conclude that coherence is preserved on timescales larger than the Rabi cycle.

On the same line, in the paragraph second to last we show that the coherence times may reach durations that are even longer than the laser repetition cycle of 12.5 ns (Fig. 4) and which can be controlled externally with the applied magnetic field according to Gilbert's theory.

In the revised version of the manuscript we have added a new paragraph which describes the relationship between the Rabi cycle times and the coherence times. In addition, the differences/similarities with the " π -pulse" case are now better explained in the introduction.

3) Compared to conventional experiments on Rabi oscillations, the experiments described here require additional laser pumping. It is claimed that the laser pulse acts as a pulse of a magnetic field. Why? What was the direction of the field? What are physical quantities that define the strength and the direction of this laser-induced field? Is it realistic to expect such a strength and duration of the laser-induced magnetic field? The authors hardly explain these issues. The physics of the effect of the laser pulse on the magnetic medium leaves a lot of questions raising doubts about the validity of the suggested model.

Though still under much debate (Refs. 24, 34), it is generally believed that the optical pump increases the lattice temperature thereby reducing temporarily the effective anisotropy, H_{Keff} , and the magnetization saturation, M_s (Beaurepaire et al. *PRL* **76**, 4250). Our experiments are different than the TRMOKE experiments in the sense that they allow to perturb the magnetization at any given point along the precessional trajectory. Hence, our data brings new information of the interaction with the optical pump that does not exist in TRMOKE

experiments. In our numerical simulations we first modeled the interaction with the pump as an instantaneous reduction in M_s and H_{Keff} according to the existing theories. This however was not sufficient to reproduce the measured traces. The main problem was that our calculation did not reproduce the field dependent phase profile at positive times near $t = 0$, namely, the curvature in the vertical contours of Fig. 2A appearing at times that immediately follow the pump. We see in the figure that immediately after the optical pulse was applied, the acquired phase is smaller near the resonance field at ~ 445 mT compared to higher or lower fields. This effect is understood to be related to the π - phase shift associated with the resonance response as the field is swept across the resonance conditions. However, when we applied in addition an impulse field of magnitude 60 mT and of 3 ps duration oriented in the sample plane along the \hat{y} direction (shown in Fig. 1B), the measured phase response around $t = 0$ was better reproduced (Fig. S3). This field was introduced phenomenologically and was able to account for our observations.

This field could originate in various effects. A plasma effect induced by the optical pump is one possibility. Although the inverse Faraday effect (Stanciu et al., PRL **99**, 047601) is responsible for the generation of a perpendicular magnetic field and is generated only as long as the optical pulse exists, it may also be the source of the observed effect. The impulse strength of 60 mT is indeed reasonably achievable by the inverse Faraday as seen from the study of Cornelissen et al. (APL **108**, 142405) who suggested that the generated field could be even as large as 20 Tesla for much more intense optical fields than in our experiments. Recently, Ellis et al. (Scientific Reports, **6**, 30522 (2016)) proposed a new theory that was based on thermal excitation above the anisotropy field together with a polarization dependent absorption and could also possibly stand behind our observations. Further experimental study of the interaction with the optical pump could benefit from carrying out our experiments on other material systems but is beyond the scope of our manuscript.

Despite the fact that the interaction with the optical pump is a topic of much debate and discussion, for the purpose of our work this uncertainty plays only a minor role. Regardless of its physical origin, the important outcome is in perturbing the steady precessional magnetization state and thereby triggering the non-adiabatic regime.

In the revised version of the manuscript a new paragraph was added that describes in more detail the considerations related to the additional pulsed magnetic field, its geometrical orientation, as well as a discussion of the possible mechanisms of excitation.

We thank the referee for a thorough examination of our manuscript and for his/her comments.

Reviewer #5 (Remarks to the Author):

The authors study Rabi oscillations in a thin-film ferromagnet by driving magnetization precession with a microwave field and perturbing the system with femtosecond laser pulses. They show that the perturbation leads to a modulation of the magnetization precession amplitude. The characteristic frequency of this modulation is found to be consistent with a generalized Rabi formula. Further, they investigate how the laser pulses

influence the phase of the oscillating signal (as function of the applied magnetic field) and how the Rabi frequency depends on the magnitude of the microwave field. They also study interference effects between subsequent responses to a train of laser pulses.

The topic of this work is innovative and the results are of broad interest. I also believe that the presented data are technically sound. The observed oscillations (with a signature consistent with Rabi oscillations) have not been reported before in ferromagnets and the results give valuable insights into the interaction between laser pulses and magnetization (e.g., an improve understanding of optical magnetization switching and how to model the interaction phenomenologically). In particular, I believe physicists working within spintronics will find this article useful for their future works. However, I think the authors overstretch the link to quantum computation, because they consider a classical system and a large ensemble of spins (not a quantum mechanical two-level system). Another weak point of the paper is its presentation. It is difficult to get a good understanding of this work without reading the article a couple of times. For instance, I miss a better introduction that gives a simple physical picture of the effect.

In conclusion, the results are novel, technically sound, and will be important to scientists working within spintronics. Therefore, I believe this paper satisfies the criteria set for being publishable in Nature Communications and recommend publication after the authors have improved the introduction with a simple physical picture of the observed Rabi oscillations.

The introduction of our paper was revised to be more coherent. That includes changing the order of the presented ideas as well as adding explanations throughout. Furthermore, several phrases that might still have caused confusion between the classical and quantum systems have been removed. With these modifications we believe that the distinction between that quantum limit and our classical ensemble-averaged system is made more apparent and that the physical picture is more easily understandable.

We thank the referee for reviewing our manuscript and for this comment.

Reviewers' comments:

Reviewer #4 (Remarks to the Author):

Although the authors provided a very detailed response, I am afraid that my criticism has not been addressed fully. In my previous report I raised three issues.

1. The first and the second issues raise concerns that the observed phenomenon requires an explanation from quantum mechanics. I am not convinced by the arguments of the authors about the quantum nature of the observed phenomenon. My concerns are also shared by Referee 5 who points out that the authors "overstretch the link to quantum computation, because they consider a classical system and a large ensemble of spins (not a quantum mechanical two-level system)."

I also notice that there is a mismatch in our definitions of the Rabi oscillations. The authors state: "The responses are however describable by a beating effect of the natural decaying precessional response at the frequency of... with the driven steady state response at the frequency of ... and is the central idea behind Rabi oscillations." According to the definition I use, the Rabi oscillations must be observed even if these two frequencies are equal (see, for instance, https://en.wikipedia.org/wiki/Rabi_frequency). Thus the beatings cannot be the central idea of the Rabi oscillations.

The authors state: "Furthermore, the numerical model we use was calculated in the macrospin limit and had no spatial resolution. Hence, it did not account for spin waves." Here I clearly see a misunderstanding, because in my report I meant the spin waves in the center of the Brillouin zone. Those waves must be taken into account in the macrospin limit.

I am also very much confused by the statement of the authors that the Gilbert damping coefficient α can be considered as a combination of T_1 and T_2 . I cannot agree. The authors should provide a better proof/explanation of such a statement and not just a reference.

2. The third concern was about the lack of information about the effect of the laser pulse on the studied material. I am quite puzzled by the response of the authors.

a) Ref. 34 is inadequate citation. The reference discusses a possibility of probing ultrafast laser-induced spin dynamics with light and not the effect of light on magnetic materials.

b) Ref. Stanciu et al., PRL 99, 047601 is not about the inverse Faraday effect (and thus inadequate), but about all-optical magnetization reversal. It is suggested that the inverse Faraday effect may play a role there, but it does not prove that such a field is really present.

c) Ref. APL 108, 142405 is inadequate citation. The paper studies a possibility of all-optical magnetization reversal under assumption that light generates an effective magnetic field. The work does not prove that such a field is really present.

d) Ellis et al. (Scientific Reports, 6, 30522 (2016)) is also cited inadequately. The paper considers a mechanism of all-optical magnetization reversal under assumption that light generates a magnetic field. The work does not prove that such a field is really present.

From the manuscript it is absolutely unclear why the light generates this effective magnetic field and why the life time of the field is so long 3 ps. I could not find any information on the polarization of the pump pulse used by the authors. It is not clear if the parameters of the experiment (pump polarization, symmetry of the sample) can allow a generation of an effective magnetic field in the required direction at all. The only evidence provided by the authors is that an existence of such a field would explain the results. I guess it is a rather weak point of the manuscript.

To summarize, it is obvious that the mutual misunderstandings between the authors and me hamper a constructive discussion. At this point I cannot agree with the authors. The authors must improve the discussion of the origin of the magnetic field generated by light. At the same time, the results are of high quality and interesting. I understand that the discussion "quantum-or-classical" will only affect the introduction of the paper and will not change the main results. Therefore, a publication of this manuscript with an improved discussion of light-matter interaction, the referee

reports and responses of the authors (as supplementary) can be very useful and interesting to magnetic society.

Reviewer #5 (Remarks to the Author):

The response of the authors to my previous comments is satisfactory.
I have no further comments and recommend publication in Nature Communications.

We thank referee #3 and #5 for their positive comments and for recommending acceptance of our paper for publication in Nat. Commun.

We address below the comments from Referee #4 (which we have highlighted in bold)

Reviewer #4 (Remarks to the Author):

We thank the referee for his/her comments which serve to improve our paper. We address hereby the referee's remarks:

Although the authors provided a very detailed response, I am afraid that my criticism has not been addressed fully. In my previous report I raised three issues.

1. The first and the second issues raise concerns that the observed phenomenon requires an explanation from quantum mechanics. I am not convinced by the arguments of the authors about the quantum nature of the observed phenomenon. My concerns are also shared by Referee 5 who points out that the authors “overstretch the link to quantum computation, because they consider a classical system and a large ensemble of spins (not a quantum mechanical two-level system).”

Following referee #5's earlier comments on our original submission, we have further modified the manuscript to make a clear distinction between the quantum and classical regimes and the relations between the two. Referee #5 concluded our first set of revisions to be satisfactory. Reading reviewer #4's comment above, we have once more revisited the introduction and made changes that further emphasize the fact that we are dealing with an ensemble of spins.

I also notice that there is a mismatch in our definitions of the Rabi oscillations. The authors state: “The responses are however describable by a beating effect of the natural decaying precessional response at the frequency of... with the driven steady state response at the frequency of ... and is the central idea behind Rabi oscillations.” According to the definition I use, the Rabi oscillations must be observed even if these two frequencies are equal (see, for instance, https://en.wikipedia.org/wiki/Rabi_frequency). Thus the beatings cannot be the central idea of the Rabi oscillations.

We wish first to clarify the mathematical-physical background.

It is well known that the Rabi problem can be represented qualitatively by the driven, damped, harmonic oscillator problem (as explained in https://en.wikipedia.org/wiki/Rabi_problem). The equation describing the harmonic driven damped oscillator problem has the following form:

$$m\ddot{y} + b\dot{y} + ky = F \cos(\omega t) \quad , \quad \omega_0 = \sqrt{\frac{k}{m}} \quad ,$$

whose solution is given by the sum of the solution to the homogeneous equation, y_h , and a particular solution of the inhomogeneous equation, y_p , so that

$$y = y_h + y_p.$$

y_h is the *transient solution* and decays exponentially in our case, while y_p is known as the steady state response and “survives” after y_h has decayed.

For simplicity, without loss of generality, we consider the non-dissipative case, for which $b = 0$. In the **non-resonant** case, $\omega_0 \neq \omega$, y_h takes the form: $y_h = c_1 \cos(\omega_0 t) + c_2 \sin(\omega_0 t)$ which oscillates at the natural frequency of the system (c_1 and c_2 are merely coefficients) while $y_p = \frac{F}{m(\omega^2 - \omega_0^2)} \cos(\omega t)$ oscillates at the driving frequency. The total solution is the sum $y_h + y_p$ and is a beating of a signal at the natural frequency, ω_0 , together with a signal at the driving frequency, ω . This solution holds only for the off-resonance case and explains intuitively the measured chirp profiles in our experiment (Fig. 2D).

For the on-resonance $\omega_0 = \omega$ case with which the referee is concerned, the problem must be solved separately and the solution can be shown to have the form, as follows:

$$y_h = c_1 \cos(\omega_0 t) + c_2 \sin(\omega_0 t) \quad \text{and} \quad y_p = \frac{F}{2m\omega_0} t \sin(\omega_0 t),$$

so that the total solution, $y = y_h + y_p$ is still a beating of two signals at the same frequency. Note that in this dissipationless case, the increase in amplitude of y stems from the particular solution, y_p . Hence, there is no contradiction with the referee’s statement that **“the Rabi oscillations must be observed even if these two frequencies are equal”**.

The behavior of our problem where a spin system resides in a magnetic field and has the natural frequency of μH_0 is in principle not much different from the behavior of the harmonic oscillator.

The complete solution for our case was described in the paper by Torrey (Phys. Rev. **76**, 1059) for the **rotating frame** for any one component of the magnetic moment and has the same form as discussed above:

$$Ae^{-a\tau} + Be^{-b\tau} \cos s\tau + \frac{C}{s} e^{-b\tau} \sin s\tau + D$$

(Eq. (4), Torrey PR **76**, 1059)

with the first three terms giving the transient effect and D being the particular solution ($a, b, s, A, B, C,$ and D are constants and τ corresponds to time in the rotating frame). Furthermore, Torrey also describes the resonant case, $\omega_0 = \omega$, and applies the same “beating formalism” of the transient solution with the steady state solution at the driving frequency.

Following the referee's comment, we have inspected again the manuscript. Consequently, we replaced the statement: "Rabi oscillations can be regarded as a beating of the natural decaying..." that might have caused confusion with a new statement that better conveys that the beating picture is useful as an intuitive description.

We find the beating picture to be useful for the broad readership in explaining intuitively the results of Fig. 2D for which the chirp profile is shown to be dependent on the magnitude of $\omega_0 = \gamma\mu H_0$ relative to ω_{rf} .

The authors state: "Furthermore, the numerical model we use was calculated in the macrospin limit and had no spatial resolution. Hence, it did not account for spin waves." Here I clearly see a misunderstanding, because in my report I meant the spin waves in the center of the Brillouin zone. Those waves must be taken into account in the macrospin limit.

Reading the referee's comment we regret to say that we did not understand his/her comment. The spin waves in the center of Brillouin zone (Gamma point) correspond to the spin waves with $k = 0$. This $k = 0$ mode corresponds to the uniform excitation mode. Hence, we do not see a difference between $k = 0$ mode and macrospin limit which we have calculated.

I am also very much confused by the statement of the authors that the Gilbert damping coefficient α can be considered as a combination of T1 and T2. I cannot agree. The authors should provide a better proof/explanation of such a statement and not just a reference.

The T_1 and T_2^* relaxation times defined in the Bloch-Bloembergen representation [Bloch PR **70**, 460, Bloembergen PR **73**, 679] describe the spin-lattice and the transverse relaxation times, respectively, with the latter being responsible also for the inhomogeneities, as defined by their equations:

$$\begin{aligned}\frac{dM_{x,y}}{dt} &= \gamma(\vec{M} \times \vec{H})_{x,y} - \frac{M_{x,y}}{T_2^*} \\ \frac{dM_z}{dt} &= \gamma(\vec{M} \times \vec{H})_z - \frac{M_z - M_0}{T_1}\end{aligned}$$

in which M_i is the component of the magnetization along the i coordinate and M_0 is the steady state magnetization along the z axis.

Alternatively, the relaxation processes can be described using the Gilbert damping coefficient, α , which takes into account the spin-lattice and transverse decays simultaneously, as used in the Landau-Lishitz-Gilbert equation:

$$\frac{d\vec{M}}{dt} = -\gamma\vec{M} \times H + \alpha \frac{1}{M_s} \vec{M} \times \frac{d\vec{M}}{dt},$$

while the inhomogeneous broadening is modeled through the variation in H_{Keff} , ΔH_{Keff} , as in Shaw et al. APL **105**, 062406, Iihama et al., PRB **89**, 174416, for example.

Graphically, T_1 , T_2^* , and Gilbert damping relaxation torques can be represented in the following manner:

The graphical representation shows that α is a nontrivial time dependent combination of T_1 and T_2^* . Eventually, the Gilbert damping was adopted by the magnetism community due to the ability to rigorously derive it from a friction term in the corresponding generalized coordinates space (Gilbert, Magnet., IEEE Trans. **40**, 3443 (2004)).

Per the referee's comment, this discussion was added to the supplementary materials while the relevant references were added to the body of the revised manuscript.

2. The third concern was about the lack of information about the effect of the laser pulse on the studied material. I am quite puzzled by the response of the authors.

- a) Ref. 34 is inadequate citation. The reference discusses a possibility of probing ultrafast laser-induced spin dynamics with light and not the effect of light on magnetic materials.
- b) Ref. Stanciu et al., PRL **99**, 047601 is not about the inverse Faraday effect (and thus inadequate), but about all-optical magnetization reversal. It is suggested that the inverse Faraday effect may play a role there, but it does not prove that such a field is really present.
- c) Ref. APL **108**, 142405 is inadequate citation. The paper studies a possibility of all-optical magnetization reversal under assumption that light generates an effective magnetic field. The work does not prove that such a field is really present.
- d) Ellis et al. (Scientific Reports, **6**, 30522 (2016)) is also cited inadequately. The paper considers a mechanism of all-optical magnetization reversal under assumption that light generates a magnetic field. The work does not prove that such a field is really present. From the manuscript it is absolutely unclear why the light generates this effective magnetic field and why the life time of the field is so long 3 ps. I could not find any information on the polarization of the pump pulse used by the authors. It is not clear if the parameters of the experiment (pump polarization, symmetry of the sample) can allow a generation of an effective magnetic field in the required direction at all. The only evidence provided by the

authors is that an existence of such a field would explain the results. I guess it is a rather weak point of the manuscript.

The pump pulses were linearly polarized and applied at an incident angle of ~ 22 degrees measured from the normal to the sample plane. The polycrystalline CoFeB thin film has a body centered cubic crystal structure that is highly non-textured hence effectively the polar coordinates are randomly oriented. Interfaces of the structure and boundaries of the crystallites naturally reduce the symmetries.

In the revised version of the manuscript, reference 34 was removed. The other references cited in the context of the inverse Faraday effect indeed describe this possibility with a varying level of certainty. Nevertheless, they are consistent in their conclusion that the optical field induces an effective magnetic field as our simulations suggest. Indeed, as the referee points out, the better agreement with the experiments obtained when the transient field is introduced does not guaranty existence or uniqueness of the field. This point is explained now in the manuscript.

Per the referee's suggestion, in order to better address the light-matter interaction, the discussion related to the interaction of the optical pulse with the ferromagnet was expanded and parts of it were moved to the supplementary materials where it appears in a new separate section of its own. This section was written in a self-explanatory manner which we believe the reader will find useful in providing insight to this controversial topic. The inverse Faraday effect is now cited in the context of the broader manifold of effects that may result in the generation of effective fields and emphasis was given that it is merely a possible source. Among the additional effects that are now described are the photo-induced anisotropy as well as the magnetic circular dichroism.

Our measurements provide new data of the interaction but at this point do not resolve the controversy. We hope that a thorough study of the interaction at $t = 0$ using the experimental technique described in our paper will provide a deeper understanding in the future. Nevertheless, the important aspect of the interaction with the pump pulses for the purpose of our work is that it eventually triggers the nonadiabatic regime.

The experimental parameters are indicated now in the revised version of the manuscript.

To summarize, it is obvious that the mutual misunderstandings between the authors and me hamper a constructive discussion. At this point I cannot agree with the authors. The authors must improve the discussion of the origin of the magnetic field generated by light. At the same time, the results are of high quality and interesting. I understand that the discussion "quantum-or-classical" will only affect the introduction of the paper and will not change the main results. Therefore, a publication of this manuscript with an improved discussion of light-matter interaction, the referee reports and responses of the authors (as supplementary) can be very useful and interesting to magnetic society.

We thank the referee for his/her comments which aid to clarify and better communicate our work.

Reviewer #5 (Remarks to the Author):

The response of the authors to my previous comments is satisfactory. I have no further comments and recommend publication in Nature Communications.

We thank the referee for reviewing our manuscript and for this comments.

REVIEWERS' COMMENTS:

Reviewer #4 (Remarks to the Author):

The revised version of the manuscript and the response letter of the authors have resolved most of misunderstandings and confusions which arose during previous reviews. Although I still have few concerns, the paper requires only minor revision. If the authors meet the remaining criticism, I am happy to recommend this paper for publication.

1. Quantum or classical.

From the response letter of the authors I have realized that the observed phenomenon is a classical analogue of the Rabi oscillations. I am also happy that the authors agree that (classical) spin-waves at the Gamma point correspond to uniform excitation mode. The mode can be modeled by the LLG equation. This is exactly what the authors have done to reproduce the observed behavior and reveal that light acts as a magnetic field.

To avoid further misunderstandings, the authors should clearly emphasize also in the manuscript that they report about a classical analogue of the Rabi oscillations.

The statement of the authors "these traces do not stem from classical spin wave interference" (line 96-96) is confusing and must be removed.

2. Relaxation times and Gilbert damping

I appreciate the detailed explanation of the definitions of the relaxation times T_1 , T_2 and the Gilbert damping given by the authors. I agree that there is a correlation between T_1 , T_2 and α . In the present version of supplementary material the authors state that the ratio is "nontrivial and time-dependent". I fully agree. This is exactly what I meant in my previous report. I just would like to note that in the experiment discussed by the authors, the femtosecond laser-pulse must cause a subpicosecond demagnetization of the material followed by a nanosecond recovery. Such demagnetization and recovery correspond to longitudinal dynamics of the magnetization. I am not sure that just the LL-equation with a Gilbert-damping is able adequately describe the laser-induced dynamics. However, the LLG equation is sufficient for the description, if the demagnetization is small and the experiment is mainly sensitive to the transverse component. I advise the authors to comment on the applicability of the LLG in the main text of the manuscript.

3. Mechanism of the laser-spin interaction.

The authors have considerably improved the discussion of the mechanism. I only would like to add that laser-induced demagnetization, which is very likely in the experiment, can also induce an effective magnetic field by changing the demagnetizing field and shape anisotropy (Phys. Rev. Lett. 88, 227201 (2002), Chemical Physics 318, 137-146 (2005)).

4. Minor comment.

The authors in Ref. 33 are missing

We thank referee for the recommending publication of our paper. We would like also to thank for the thorough examination, and for the comments which aid to convey our results to the reader.

We address below referee #4's comments (which we have highlighted in bold)

Reviewer #4 (Remarks to the Author):

The revised version of the manuscript and the response letter of the authors have resolved most of misunderstandings and confusions which arose during previous reviews. Although I still have few concerns, the paper requires only minor revision. If the authors meet the remaining criticism, I am happy to recommend this paper for publication.

1. Quantum or classical.

From the response letter of the authors I have realized that the observed phenomenon is a classical analogue of the Rabi oscillations. I am also happy that the authors agree that (classical) spin-waves at the Gamma point correspond to uniform excitation mode. The mode can be modeled by the LLG equation. This is exactly what the authors have done to reproduce the observed behavior and reveal that light acts as a magnetic field. To avoid further misunderstandings, the authors should clearly emphasize also in the manuscript that they report about a classical analogue of the Rabi oscillations. The statement of the authors “these traces do not stem from classical spin wave interference” (line 96-96) is confusing and must be removed.

To further emphasize the fact that we report the classical analogue of Rabi oscillations, we have included now in the abstract the following sentence: “Using a combination of femtosecond laser pulses and microwave excitations, we report the classical analogue of Rabi oscillations in ensemble-averaged spins of a ferromagnetic system”. This modification should further prevent any misunderstandings that might occur.

The statement “**these traces do not stem from classical spin wave interference**” has been modified to avoid the confusion. The word “classical” was removed. This change should resolve the confusion. Other instances were corrected as well. We thank the reviewer for pointing this out.

2. Relaxation times and Gilbert damping

I appreciate the detailed explanation of the definitions of the relaxation times T1, T2 and the Gilbert damping given by the authors. I agree that there is a correlation between T1, T2 and alpha. In the present version of supplementary material the authors state that the ratio is “nontrivial and time-dependent”. I fully agree. This is exactly what I meant in my previous report.

I just would like to note that in the experiment discussed by the authors, the femtosecond laser-pulse must cause a subpicosecond demagnetization of the material followed by a nanosecond recovery. Such demagnetization and recovery correspond to longitudinal dynamics of the magnetization. I am not sure that just the LL-equation with a Gilbert-damping is able adequately describe the laser-induced dynamics. However, the LLG equation is sufficient for the description, if the demagnetization is small and the experiment

is mainly sensitive to the transverse component. I advise the authors to comment on the applicability of the LLG in the main text of the manuscript.

We agree with the referee's comment. It is possible that at lower optical pump powers the LLG equation may result in a better experimental description. We have added in the relevant paragraph a comment stating that it is possible that at lower optical pump intensities the Landau-Lifshitz-Gilbert equation may solely account for the experimental results.

3. Mechanism of the laser-spin interaction.

The authors have considerably improved the discussion of the mechanism. I only would like to add that laser-induced demagnetization, which is very likely in the experiment, can also induce an effective magnetic field by changing the demagnetizing field and shape anisotropy (Phys. Rev. Lett. 88, 227201 (2002), Chemical Physics 318, 137–146 (2005)).

Indeed changes in the demagnetization and anisotropy fields may have an important role in generation of the effective field. This important possible explanation is now described in the revised version of the manuscript including in the supplemental materials. The corresponding references suggested by the referee were added as well.

4. Minor comment.

The authors in Ref. 33 are missing

The complete authors list has been added to Ref. 33.

We kindly thank the referee for his/her remarks.